# Strategies to Build Hybrid Protein–DNA Nanostructures

**DOI:** 10.3390/nano11051332

**Published:** 2021-05-18

**Authors:** Armando Hernandez-Garcia

**Affiliations:** Laboratory of Biomolecular Engineering and Bionanotechnology, Instituto de Química, Universidad Nacional Autónoma de México, Ciudad Universitaria, Ciudad de México 04510, Mexico; armandohg@iquimica.unam.mx

**Keywords:** DNA nanotechnology, protein nanotechnology, self-assembly, bionanomaterials

## Abstract

Proteins and DNA exhibit key physical chemical properties that make them advantageous for building nanostructures with outstanding features. Both DNA and protein nanotechnology have growth notably and proved to be fertile disciplines. The combination of both types of nanotechnologies is helpful to overcome the individual weaknesses and limitations of each one, paving the way for the continuing diversification of structural nanotechnologies. Recent studies have implemented a synergistic combination of both biomolecules to assemble unique and sophisticate protein–DNA nanostructures. These hybrid nanostructures are highly programmable and display remarkable features that create new opportunities to build on the nanoscale. This review focuses on the strategies deployed to create hybrid protein–DNA nanostructures. Here, we discuss strategies such as polymerization, spatial directing and organizing, coating, and rigidizing or folding DNA into particular shapes or moving parts. The enrichment of structural DNA nanotechnology by incorporating protein nanotechnology has been clearly demonstrated and still shows a large potential to create useful and advanced materials with cell-like properties or dynamic systems. It can be expected that structural protein–DNA nanotechnology will open new avenues in the fabrication of nanoassemblies with unique functional applications and enrich the toolbox of bionanotechnology.

## 1. Introduction

The versatility and programmability of DNA on the nanoscale has been demonstrated by the notable growth and diversification that structural DNA nanotechnology has displayed in the last decades [1,2,3,4]. The continues growth and diversification of structural DNA nanotechnology has been paved by the establishment of novel strategies to build DNA structures [3,5]. For example, the building of DNA-junctions and DNA-crossovers [1], pioneered by Seeman in the 1980s, and DNA origami [6], developed by Rothemund in the early 2000s, were important milestones. These building blocks served as the foundation to build novel, complex, and hierarchical nanostructures that inaugurated new subfields in the genealogy of structural DNA nanotechnology [3]. 

The scope of structural DNA nanotechnology has been further expanded by the incorporation of other type of building blocks [3]. For example, the incorporation of inorganic nanoparticles by Mirkin and his collaborators lead to the development of programmable DNA-based colloidal crystals (commonly referred as spherical nucleic acids) with applications in photonics, electronics, and self-assembly [5]. These hybrid nanomaterials combine ssDNA molecules with rigid templates made up of inorganic nanoparticles. The latter acts as the brick that organizes and dictates the shape, while the former works as “glue” by establishing directional “bonds”. DNA-based colloidal crystals are an early example of how the combination of DNA with other types of building blocks could generate a whole new area of programmable hybrid nanomaterials.

Similarly, the incorporation of proteins into structural DNA nanotechnology as a cobuilding block further expanded the scope of DNA nanotechnology and sprouted new research avenues [7,8]. The use of proteins in DNA nanotechnology is indeed not new. Proteins have been implemented since the dawn of structural DNA nanotechnology [9,10]. Nevertheless, the accumulated advances in the understanding of protein self-assembly, design, and engineering has increased interest in integrating them into DNA nanotechnology. However, until now, the incorporation of proteins has mostly been limited to equipping DNA nanostructures with specific functionalities, for example, molecular recognition [11] or catalytic activity [12]. This limited use of proteins contrast with the myriad functions and capabilities that nucleoprotein complexes perform in nature (e.g., genetic switches, ribosomes, nucleosomes, and viruses). Looking at the large structural and functional diversity of nucleoproteins, we can appreciate the potential that proteins have in terms of working synergically with DNA building blocks.

More recently, the incorporation of proteins to control or enhance the structural features or properties of DNA nanostructures has been implemented; however, it still remains largely unexplored. This review focuses on the structural roles that proteins offer for building hybrid nucleoprotein nanostructures through their combined self-assembly with diverse DNA building blocks (e.g., DNA origami, DNA junctions, plasmid, linear DNA). We review seminal and recent work to show the strategies deployed to build hybrid protein–DNA nanostructures. We demonstrate how the full integration of proteins into DNA nanotechnology, mainly through structural and mechanical roles, makes it possible to build remarkable and unique nanomaterials and exploit all the potential benefits that these hybrid materials can offer. In order to review the application of the nonstructural roles of proteins in DNA nanostructures (e.g., the arrangement of proteins or enzymes on preassembled DNA nanostructures) as well as the structural roles that DNA offers in building nanostructures, we suggest exploring other recently published reviews [4,7,8,13]. 

## 2. Structural Protein–DNA Nanotechnology

In comparison to DNA, proteins display more complex, sophisticate, and diverse functions, as well as a larger structural diversity. These include, for example, highly specific catalytic activity, potent molecular recognition, tight and precise allosteric regulation, efficient cargo encapsulation, responsive structural functions, and cooperative binding. Since the early developments in structural DNA nanotechnology, these functionalities have been harnessed to increase the functionality of DNA nanomaterials. They provide advanced functionalities such as enhanced recognition for cellular ligands or enzymatic cascades. To achieve this, proteins are precisely positioned on a previously assembled DNA nanostructure (e.g., DNA origami or DNA junctions) (Figure 1a) [9,10,11,14,15]. Moreover, by adding functional capabilities to otherwise inert DNA nanostructures, proteins can cooperate synergistically and bring important features to the final assembled hybrid nanostructure (Figure 1a,b). 

Two distinctive approaches to how proteins are combined with DNA can be clearly distinguished: (1) proteins for function and (2) proteins for structure (Figure 1a). The first represents “functional Protein DNA nanotechnology”, whereas the second “structural Protein-DNA nanotechnology”. This review focuses on the second approach, which we refer to in the review as “structural protein-DNA nanotechnology”. In this hybrid protein–DNA nanotechnology, proteins and DNA act synergistically during the self-assembly process and serve as the foundation for the final nanostructure. Meaning that protein and DNA nanotechnologies show a high degree of structural integration.

By advantageously harnessing the biophysical and chemical properties from both biomolecules [16], structural protein–DNA nanotechnology has reduced the limitations that each molecule present when used alone. Proteins have a larger chemical and structural diversity than DNA and, although proteins alone can build sophisticated nanostructures, their versatility and programmability are severely limited due to intricate sequence–structure relationships. On the other hand, DNA lacks the ample structural and chemical diversity seen in proteins but has more predictable folding than proteins due to the readily programmable Watson–Crick interactions. Since protein–DNA nanotechnology aims to harness the different but highly complementary physical–chemical and structural properties of both biomolecules, their synergistic combination offers strategical benefits for the fabrication of nanomaterials. 

Structural protein–DNA nanotechnology is different from other common uses of proteins in DNA nanostructures because proteins play important structural, mechanical, and/or assembling roles. Although both DNA and proteins provide these roles, their degree of participation depends on the structural complexity of the starting and final structure. However, as it is shown below, most of the literature shows that the use of proteins is more operative than DNA. It is considered that proteins and DNA have structural roles in a particular protein–DNA nanostructure when it is not possible to achieve such a final nanostructure without the coparticipation of both building blocks (Figure 1a). Hence, the nonexistence of one building block does not lead to the acquisition of a particular shape, size, order, organization, or certain mechanical or dynamic properties. This means that the removal or the absence of one of them (protein or DNA) disassembles the structure or largely compromises its stability or properties.

As a result of their large structural and chemical diversity, proteins can bring multiple advantages when used for structural purposes (Figure 1b). They can spatially align DNA in specific geometries and preserve DNA topology by coating and stiffening. Furthermore, proteins can establish strong and specific interactions with ssDNA, and in particular, with DNA duplexes. This opens the opportunity to incorporate dsDNA into current DNA nanotechnology, which in turn relies on ssDNA (M13 virus plasmid and staple oligonucleotides) [17,18]. As proteins offer the advantage of working isothermally and at environmental temperatures, they can reduce the dependence on DNA molecules and multitemperature assembly processes of DNA nanotechnology. Therefore, proteins have a large potential to significantly reduce the production costs and simplify assembly processes, which currently limits the large-scale use of DNA nanotechnology in many applications.

## 3. Proteins in Hybrid Nanotechnology

In order to form nanostructures with DNA, proteins need to establish strong and effective interactions with DNA building blocks. Several types of proteins have been used in structural hybrid nanotechnology. These include enzymes, multimeric proteins, metal-binding proteins, coiled-coil peptides, cationic peptides, cationic polymer proteins, ribosomal proteins, transcription factors, viral proteins, nucleosomes, polymerases, and others (Table 1). These proteins interact with DNA through two different approaches: (1) covalent conjugation or (2) noncovalent coassembly (Figure 2). Covalent conjugation is usually performed by chemically linking proteins and DNA through reactive groups (Figure 2a). The DNA can be an assembled nanostructure or oligos with complementary sequences. Covalent conjugation is frequently used because it is straightforward, and it is easy to control and render (bio)chemically stable conjugates [13]. This strategy also means that practically any protein carrying the proper reactive group can be conjugated. On the other hand, noncovalent coassembly requires using proteins with DNA-binding capabilities (Figure 2b). Since noncovalent interactions are tunable and reversible, they offer the possibility to create flexible, modular, and highly dynamic hybrid nanostructures with advanced and complex functionalities that can mimic natural nucleoprotein complexes. However, the resultant complexes can have low stability and be more susceptible to the environmental conditions than chemically linked complexes; thus, the control of these types of interaction represents a great challenge. 

The reported proteins (lacking DNA-binding capabilities) conjugated covalently to DNA include (multimeric) enzymes used as structural templates [19,20,21], metal-binding proteins [22], coiled-coil peptides [23], and elastine-like peptides [24]. By contrast, proteins that exhibit a DNA binding affinity in a sequence-dependent or independent mode include cationic peptides [25,26,27,28], cationic polymer proteins [29,30,31], ribosomal proteins [32], transcription activator-like (TAL) effectors [17], transcription factors [33], viral proteins [34,35,36], histones, and polymerases [37]. Simpler options such as streptavidin or bioinspired cationic protein polymers made up of extremely simple and repetitive amino acids that retain DNA-binding functionality or even virus-like properties have also been used [7,25,38]. These proteins can be used in combination with junctions, tiles, motifs, or origamis, and also single ssDNA or dsDNA molecules.

**Table 1 nanomaterials-11-01332-t001:** Proteins used in structural protein–DNA nanotechnology.

Building Block	Type of Protein	Building Strategy	Interaction ^1^	Ref.
βGalactoside 1D-DNA conjugate	Enzyme	(1) Structural scaffold to attach DNA	C	[19]
		(2) Polymerization		
GroEL-DNA conjugate	Chaperonin	(1) Structural scaffold to attach DNA	C	[20]
		(2) Polymerization		
RIDC3-DNA conjugate	Engineered tetrameric metal-interacting cytochrome cb56	(1) Structural scaffold to attach DNA	C	[22]
		(2) Polymerization		
Drosophila Engrailed homeodomain (ENH)	Engineered transcription factor	(1) Polymerization	NC	[33]
Coiled coil-DNA conjugate	De novo dimerizing peptide	(1) Polymerization	C	[23]
K3C6SPD	Engineered self-assembly β-sheet cationic peptide	(1) Polymerization	NC	[25]
CP++ and sCP	Designed self-assembly cationic collagen mimetic peptides	(1) Polymerization	NC	[27]
Aldolase-DNA conjugate	Trimeric enzyme	(1) Structural scaffold to attach DNA	C	[21]
		(2) Spatial organization		
H2A, H2B, H3 and H4	Histone proteins forming nucleosomes (Chromatin)	(1) Spatial organization	NC	[39]
Streptavidin	Tetrameric biotin binding protein	(1) Spatial organization	NC	[40,41]
Traptavidin	Engineered tetrameric biotin binding protein	(1) Spatial organization	NC	[42]
I3V3A3G3K3	Engineered self-assembly β-sheet cationic peptide	(1) No programmable folding of DNA	NC	[28]
L7Ae	RNA-binding ribosomal protein	(1) Bending	NC	[32,43]
		(2) Conformational change		
Transcription activator–like (TAL) effector	Engineered bivalent proteins for recognition of specific DNA sequences	(1) Programmable folding of DNA	NC	[17]
RecA	DNA-binding protein involved in the repair and maintenance of DNA	(1) Self-assembly	NC	[44]
		(2) Coating		
		(3) Rigidifying		
Tobacco Mosaic Virus coat protein	Viral RNA binding protein	(1) Self-assembly	NC	[34,35]
		(2) Coating		
		(3) Rigidifying		
		(4) Dynamic systems		
Redβ	Single-strand annealing protein for homologous recombination in phages	(1) Coating	NC	[36]
		(2) Rigidifying		
C_8_-B^Sso7d^	Engineered diblock protein polymer carrying a nonsequence specific dsDNA binding domain from archeal origin	(1) Coating	NC	[18,31,45]
		(2) Rigidifying		
C_4_-S_10_-B^K12^	Engineered triblock cationic protein polymer	(1) Coating	NC	[30]
		(2) Rigidifying		
C_4_-B^K12^	Engineered diblock cationic protein polymer	(1) Coating	NC	[29,46]
		(2) Rigidifying		
T7RNAP-ZIF	Engineered T7 RNA polymerase fused to a DNA-binding zinc finger motif	(1) Moving DNA parts	NC	[37]
(GVGVP)_40_	Engineered elastin-like polypeptide	(1) Dynamic and responsive systems	C	[24]

^1^ Tables may have a footer. C: Covalent conjugation, NC: Noncovalent interaction.

## 4. Strategies to Build Protein–DNA Nanostructures

In structural protein–DNA nanotechnology, proteins and DNA building blocks have been combined following a variety of strategies. The strategies reported until now include polymerization, directing and spatial organization, bending, folding, self-assembly, coating, rigidizing, and moving DNA parts (Figure 3). In these processes, DNA and proteins act synergically to build more complex structures. However, proteins generally have a more active role during the assembly than DNA. However, DNA plays an important role in the self-assembly of the final structure by operating as a structural template or scaffold for protein binding or anchoring. Below, it is discussed the aforementioned strategies.

### 4.1. Polymerizing DNA

Perhaps the most basic strategy is to bring together DNA and protein components to polymerize them (Figure 4). Polymerization occurs through complementary DNA oligos attached to proteins or through protein–protein interactions. Proteins that polymerize with themselves and with DNA either by chemical conjugation to self-complementary oligos or through specific interactions have been used to create one-dimensional protein–DNA nanomaterials. A pair of dimerizing coiled coils were chemically conjugated to ssDNA oligonucleotides complementary to oligos located on the surface of a DNA origami (Figure 4a) [23]. When the DNA-coiled coil conjugates self-assemble into antiparallel dimers, DNA origamis were brought together, and elongated megadalton-size nanostructures were obtained. Other reports followed similar approaches. For example, protein dimers that coassemble through metal-directed protein–protein self-assembly were further polymerized through complementary ssDNA strands (Figure 4b) [22]. Analogously, proteins covalently conjugated to ssDNA strands positioned on opposing faces have been harnessed to create large one-dimensional nanotubes or fibrous nanomaterials (Figure 4c) [19,20,47]. 

A more sophisticated strategy to build one-dimensional nanomaterials was reported by Mayo and his team [20]. They computationally designed a protein able to establish dual protein–protein homodimerization and protein–DNA interactions (Figure 4d). They combined a DNA-binding domain with the engrail *Drosophila* homeodomain to create a protein capable of establishing both interactions on opposing sides. In the presence of dsDNA, the designed protein self-assembled in hybrid nanowires, consisting of two interactive proteins bridging dsDNA molecules on both sides. 

### 4.2. Directing and Organizing DNA in Space

Using proteins to spatially orient or arrange a number of DNA components is another common strategy reported to build hybrid nanostructures (Figure 5). A very early attempt to create directional hybrid protein–DNA complexes was achieved by Chengde Mao and his group [40]. They attached four biotynilated dsDNA fragments to a central streptavidin tetramer, resulting in a cross-shaped complex, which was, however, too flexible to build anything (Figure 5a). Later, this concept was further developed by conjugating four different short ssDNAs, instead of long dsDNA fragments, into Traptavidin, a form of chemical and mechanical-resistant streptavidin (Figure 5b) [42]. The resultant protein–DNA complex was associated with magnetic beads and semiconductor nanoparticles through complementary ssDNA strands for applications in plasmonics.

With the aim of increasing the dimensionality of hybrid nanostructures, combinations of proteins and DNA have been successful in producing 3D nanostructures. More sophisticated nanostructures than simply linear examples have been created using protein handlers. Several streptavidin tetramer proteins were grafted onto each face of DNA polyhedras and biotinylated at each wire of the frame [41]. This rendered a more complex and richer 3D protein–DNA nanostructure than the original DNA polyhedra. Another example of constructing spatially defined and tunable 3D tetrahedral cages involves the self-assembly of a homotrimeric protein, covalently joined to three identical ssDNA handles (one in each protein) to a triangular DNA base carrying complementary ssDNA strands at each corner (Figure 5c). In other example, precise and regular geometrical 3D nanostructures were built by harnessing the natural propensity of histones to form quaternary structures and specifically bind ssDNA [39]. 

### 4.3. Shaping DNA through Bending and Folding

In this strategy, proteins bring together different intramolecular parts of a nucleic acid building block to obtain a nucleoprotein complex with a regular and well-defined shape (Figure 6) [17,28,32,48]. These principles were elegantly harnessed by Dietz and his group to self-assemble remarkable DNA–protein hybrids into 2D and 3D nanoshapes (Figure 6a) [39]. They engineered dozens of TAL effector proteins, each one able to bind onto two distant intramolecular positions of a flexible linear dsDNA molecule. Upon binding onto the sites, the bivalent TAL effector proteins orchestrate the folding of the DNA molecule into a previously designed shape. The assembly of protein–DNA hybrid nanostructures was demonstrated in a cell-free system from genetic components codifying for the proteins, suggesting that such hybrid protein–DNA nanostructures could be biologically produced. In another interesting example, Hirohisa Ohno et al. used the ribosomal RNA-binding protein L7Ae to bend dsRNA molecules into geometrical shapes (Figure 6b). L7Ae bends dsRNA by tightly interacting in particular sequences called K-turns and inducing a conformational change of approximately 60° on them [32]. Using this strategy, they created a synthetic RNA–protein nanostructure shaped like an equilateral triangle. The group later extended this idea to engineer RNA–protein complexes with different nanoarchitectures and applied them to imaging and therapeutic applications [43,49]. 

### 4.4. Protein Self-Assembly on DNA

Sometimes a DNA building block is used as a foundation for protein self-assembly on its surface (Figure 7). This strategy has been exploited to render hybrid nanostructures with new shapes and properties than the original DNA building block. Linear dsDNA molecules have templated the self-assembly of RecA protein filaments [44], virus-like β-sheet forming peptides [28], and Tobacco Mosaic Virus (TMV)-inspired proteins [30]. RecA protein allows double-stranded DNA scaffolds to be patterned in a programmable and site-specific fashion (Figure 7a). A TMV-inspired protein called C-S_Q10_-B^K12^ self-assembled on dsDNA and condensed it into regular protein–DNA nanorods (Figure 7b). Other groups used DNA nanostructures instead of monomolecular DNA templates. Sophisticated nanoarchitectures were created through the in situ assembly of TMV coat proteins onto genome-mimicking RNA strands anchored to the sides of DNA nanotubes (Figure 7c) or the vertexes of DNA nanotriangles [34]. This was exploited to render hybrid nanostructures with new and different properties than the original DNA nanostructure.

### 4.5. Coating and Rigidizing DNA

When a single DNA block is coated by various copies of a protein, this results in a stiff complex (Figure 8). This simple and direct strategy can be exploited to assemble self-sustained topological protein–DNA nanostructures [18,31,36,45,50]. This happens because a DNA-binding protein increases the rigidity of very flexible DNA parts (usually double strand) in a DNA nanostructure. The protein coating reveals and sustains the previous floppy topology of the initial DNA nanostructure. The coating-and-rigidizing strategy also has the extra advantages of increasing the enzymatic and thermal stability of the protein-coated DNA building block.

A clear example of this strategy is the protein C_8_–B^Sso7d^ (Figure 8a). It consists of a colloidal stability block that is attached to a nonsequence specific DNA binding affinity domain (Sso7d, 7 kDa). The considerable stiffening effect that this DNA-coating protein has over linear dsDNA molecules has been harnessed to build star-like DNA nanostructures from linear DNA molecules [18]. C_8_–B^Sso7d^ has also been used to coat 2D and 3D DNA origamis without any observable structural perturbation. C_8_–B^Sso7d^-coated DNA origamis presented higher biochemical stability and needed lower amounts of Mg^2+^ to be assembled and improve aqueous dispersion and decoration with gold nanoparticles than naked DNA origamis [31,45]. This strategy could be used to reduce the leakage of small-sized drugs encapsulated in the interior of 3D DNA nanostructures.

Similarly, other proteins have been used to coat DNA and create hybrid nanomaterials. For example, C_4_-B^K12^, a DNA-coating protein related to C_8_–B^Sso7d^, was used to resolve fluorescent markers with high spatial resolution in single DNA molecules inside nanochannels upon their binding and stiffening [46]. RecA protein filaments have been exploited to provide rigidity to DNA wires [44,50] and to build tetrahedral nanostructures with defined dimensions (Figure 8b) [50]. Similarly, the single-strand annealing protein Redβ was used to form rigid blunt-ended four-arm junctions [36]. 

### 4.6. Dynamic Nanostructures: Moving DNA Parts

Dynamic protein–DNA systems have a large potential because they are able to carry very complex processes such as the ones observed in complex viruses or in cellular components such as motors, compartments, and ribosomes. The development of dynamic hybrid biomaterials is still in its infancy; however, the few examples reported here demonstrate the potential of this strategy. In this strategy, proteins or DNA parts actuate on the other building blocks of the system, meaning that they act as switches or molecular motors (Figure 9). Without doubt, this is one of the most advanced functions that proteins can bring to the field of hybrid protein–DNA nanotechnology. 

Pirzer and his collaborators functionalized a rectangle DNA origami with elastin-like polypeptides that could reversibly fold the nanostructure upon hydrophylic–hydrophobic transition of the ELP base on salt concentration or temperature changes (Figure 9a) [24]. Famulok and collaborators built a hybrid nanoengine by coupling an engineered zinc finger and a T7 RNA polymerase, to a circular dsDNA (stator) catenated into a smaller and rigid circular dsDNA (rotor) (Figure 9b) [37]. The protein fusion, acting as an engine anchored onto a T7 promotor located in the rotor and the polymerase, was able to move it. This particular system could help develop novel nanomaterials with dynamic capabilities.

In another outstanding study, the dynamic and step-wise assembly of TMV proteins was controlled by DNA nanostructures [35] (Figure 9c). The genome-mimicking RNA anchored on triangular or barrel origami nanostructures was locked by a series of DNA strands on the template, preventing the protein from self-assembling on the RNA. Using toehold-mediated strand displacement, the viral genome was released stepwise, leading to in situ dynamic TMV assembly and the production of a DNA−protein hybrid nanostructure. This work demonstrates that, using clever DNA strand-displacement technologies, DNA could actively participate in creating dynamic hybrid nanostructures. Furthermore, combining proteins and DNA nanostructures can generate systems reassembling the packaging mechanisms observed in viral systems, meaning that the information flow leading to protein self-assembly can be regulated using DNA nanotechnology. Although combining proteins and DNA to create successfully dynamic structures is being explored, the field is in its infancy in terms of the wide-ranging possibilities.

## 5. Functional Applications

A large and diverse list of potential applications for this relatively new type of nanomaterial has been suggested. However, instead of demonstrating their applications, researchers put considerable effort into establishing basic rules and general guidelines for fabrication. Indeed, most applications of protein–DNA nanostructures are currently in the “proof-of-concept” stage of investigation. Another interesting point is that many researchers envision their hybrid nanostructures as versatile platforms for many different applications. 

The most anticipated applications are in the fields of nanomedicine, synthetic biology, structural biology and biophysics, bioinspired nanomaterials, and nanorobotics. Within nanomedicine, the development of nanobiosensors for molecular imaging, as well as smart and responsive drug and gene delivery systems are being explored [43]. The latter is the most studied application [38,51,52]. The combination of sensing and drug delivery could lead to the creation of theragnostic hybrid materials [42]. Other applications of high relevance for nanomedicine are the creation of bioresponsive nanomaterials [24] and immunomaterials, such as multivalent vaccines. Potential applications inside structural biology and biophysics include the establishment of nanoplatforms for investigating the structure and assembly mechanisms of viruses [30,48] and controlling the architecture of genomic DNA and gene expression by looping DNA [17]. Protein–DNA nanostructures acting as scaffolds to attach multiple enzymes involved in the biosynthetic pathways [53] of high-value molecules are of particular interest in synthetic biology.

An area in which hybrid protein–DNA nanostructures have a particularly large potential is the mimicking of nucleoprotein complexes such as viruses, transcription factors, and ribosomes. Viromimetic hybrid materials have been developed into gene delivery systems and simple virus-like models for biophysical and structural studies [25,28,30,48]. With the development of dynamic protein–DNA nanostructures [37], it is possible to envision mimicking more complex structures such as ribosomes or motors for injecting DNA [35]. More advanced potential applications involve the fabrication of nanodevices, smart machines, and nanorobots. These molecular nanomachines could detect signals, localize target proteins, and control living cell function and fate [49]. Finally, nonbiological applications include the development of materials with catalytic properties, nanowires for nanoelectronics, and broader nature-inspired nanotechnology.

## 6. Perspectives

The capabilities of structural protein–DNA nanotechnology will continue developing and the repertoire of hybrid nanostructures will continue increasing. This will bring new conceptual and technical developments into the field of nanomaterials, along with new and useful applications. In order for novel hybrid nanotechnology to develop to its full potential, there are challenges that need to be addressed. One main limitation is the inherently limited knowledge about the proteins that are put to work together with DNA. In some cases, the proteins used in protein-DNA nanotechnology are complex entities, and their mechanisms are not well understood or available for fine-tuning. Another practical limitation is that the DNA nanostructure or DNA strands in the vicinity of the protein can undesirably affect the properties of the protein, making them lose their binding capabilities; or vice versa, the protein can affect the DNA structure. Highly charged proteins can change the structure of DNA and highly negatively charged proteins or hydrophobic patches can alter protein folding or conformation, causing it to lose the desired functionality. When working with DNA origami as building blocks, bulky proteins can interfere with the origami structure or with its assembly. This is true for proteins that kink or bend DNA upon binding. The physical properties of the DNA scaffold may also directly influence the activity of attached enzymes. Furthermore, the inherent asymmetry of protein surfaces can limit their application to form regular nanostructures when attaching oligos.

The design and engineering of optimized DNA-binding proteins will help advance emerging structural protein–DNA nanotechnology. For example, increasing the number of DNA-binding proteins able to work in varying conditions or making them more stable and robust should be a focus. Demonstrating practical applications of hybrid protein– DNA nanostructures will also help to consolidate this emerging field [7,13]. The use of RNA-building blocks and RNA-binding proteins has scarcely been explored. They could follow a similar path to that of DNA, whilst also opening new avenues and offering unique advantages. Hybrid DNA nanotechnology can be explored beyond proteins and complemented by integrating other biomolecules, such as RNA, carbohydrates, or lipids. This could lead to structures with cell-like functionalities.

## 7. Conclusions

Protein–DNA nanotechnology has moved beyond solely arranging proteins on the surface of DNA building blocks or nanostructures. Structural protein–DNA nanotechnology synergistically combines both molecules, usually at room temperature, to build nanosystems with unique properties not possible when using proteins or DNA independently. This hybrid nanotechnology is an amalgamation of the molecular recognition and self-assembly capabilities of proteins with the Watson–Crick base pairing programming of DNA. A diverse selection of DNA-binding proteins from natural or artificial origins have been exploited or engineered to work with DNA building blocks, which vary in nature and features, for example, monomolecular templates, such as ssDNA and dsDNA, and self-assembled structures, such as DNA tiles and DNA origami. In protein–DNA nanotechnology, various strategies have been exploited to assemble unique nanomaterials and nanoentities, for example, DNA polymerization, spatial organization and orientation of DNA, shaping and bending DNA, coating and rigidizing DNA, protein assembly on DNA, and moving DNA parts and creating dynamic structures or systems. The synergistic combination of DNA and proteins has been utilized to build highly ordered nanostructures with advanced functionalities with the potential to accomplish functions similar to their natural nucleoprotein counterparts or even surpass them. Taking full advantage of structural protein–DNA nanotechnology can greatly expand the horizons of nanotechnology.

## Figures and Tables

**Figure 1 nanomaterials-11-01332-f001:**
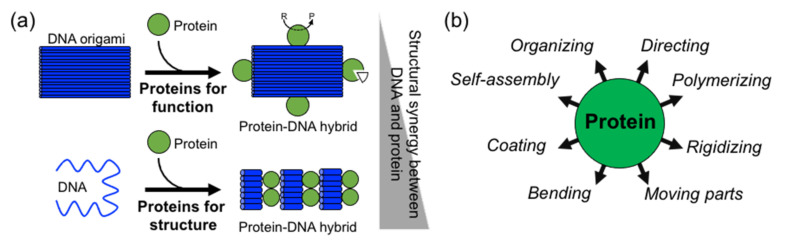
Overview of structural protein–DNA nanotechnology. (**a**) “Proteins for function” versus “proteins for structure” in DNA nanotechnology. In the latter case, there is much more synergy between the protein and the DNA building blocks. “R” means reactive and “P” product. (**b**) Structural roles of proteins in hybrid protein–DNA nanotechnology.

**Figure 2 nanomaterials-11-01332-f002:**
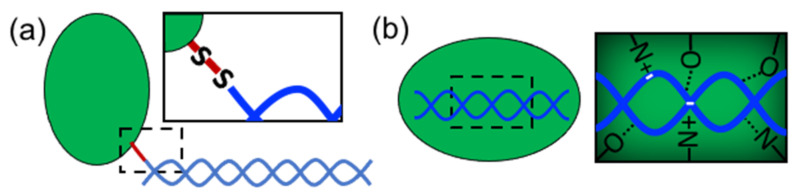
Approaches to link proteins and DNA. (**a**) Covalent conjugation (“S-S” represents a disulfide bridge) and (**b**) noncovalent interactions (dotted lines represent hydrogen bonds).

**Figure 3 nanomaterials-11-01332-f003:**
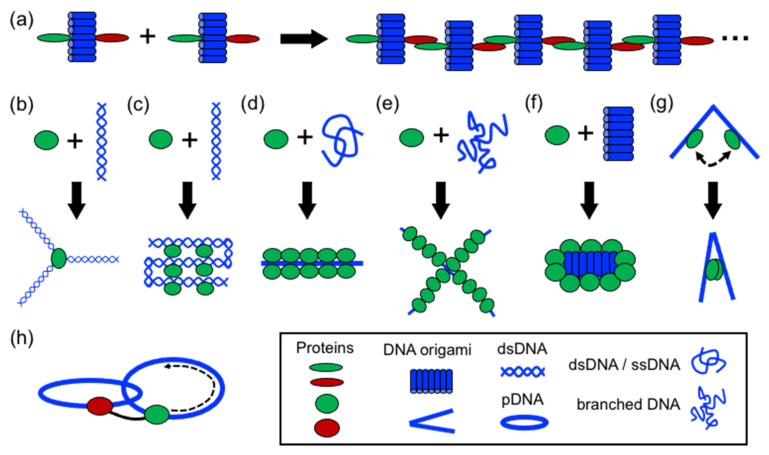
Strategies in structural protein–DNA nanotechnology. (**a**) Polymerizing; (**b**) directing spatial organization; (**c**) shaping DNA through bending and folding; (**d**) protein self-assembly on DNA; (**e**,**f**) coating and rigidizing DNA; (**g**) switching and (**h**) moving DNA components. DNA origami, ssDNA, and dsDNA molecules are depicted in blue; proteins are depicted in green and red.

**Figure 4 nanomaterials-11-01332-f004:**
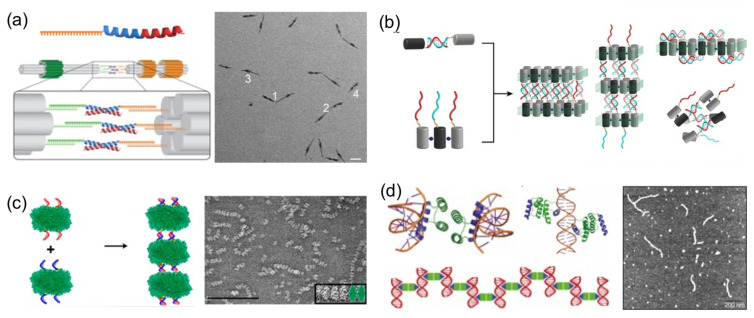
Polymerizing proteins and DNA into 1D nanostructures. (**a**) Coiled-coils conjugated to oligonucleotides (top left) dimerize DNA origamis (bottom left). TEM image is shown (right). Scale bar: 50 nm. Adapted from [23], with permission from American Chemical Society, 2019. (**b**) Metal-mediated proteins conjugated to DNA oligonucleotides self-assemble into 1D DNA assemblies. Reprinted from [22], with permission from American Chemical Society, 2018. (**c**) Enzime β-galactosidase conjugated to complementary oligonucleotides on opposing faces polymerize into elongated nanostructures (left). TEM image (right). Scale bar: 200 nm. Reprinted from [19], with permission from American Chemical Society, 2018. (**d**) Rationally designed protein with dual protein–protein and protein–DNA interactions self-assembles into nanorods (left). TEM image (right). Adapted from [33], with permission from Springer Nature, 2015.

**Figure 5 nanomaterials-11-01332-f005:**
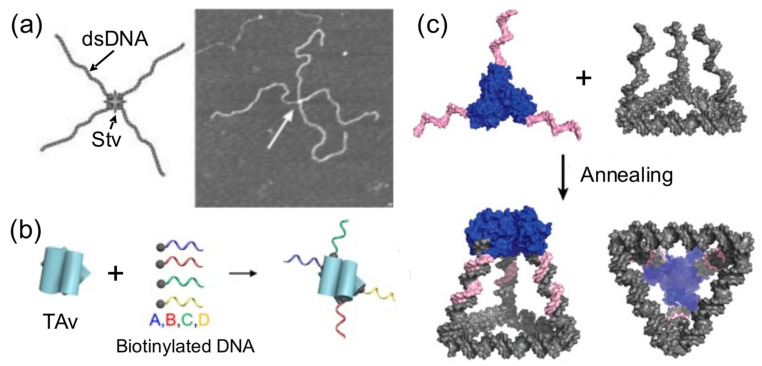
Proteins direct and organize DNA in space. (**a**) Streptavidin (Stv) tetramers binds to 4 copies of a biotinylated dsDNA (left). AFM image (right). Adapted from [40], with permission from Royal Society of Chemistry, 2006. (**b**) Traptavidin (TAv) tetramer self-assembles four biotinylated ssDNA oligonucleotides. Adapted from [42], with permission from American Chemical Society, 2019. (**c**) 3D nanocages are formed between the assembly of a triangular DNA nanostructure carrying complementary oligos in its vertices with a protein trimer conjugated to complementary oligonucleotides. Adapted from [21], with permission from American Chemical Society, 2019.

**Figure 6 nanomaterials-11-01332-f006:**
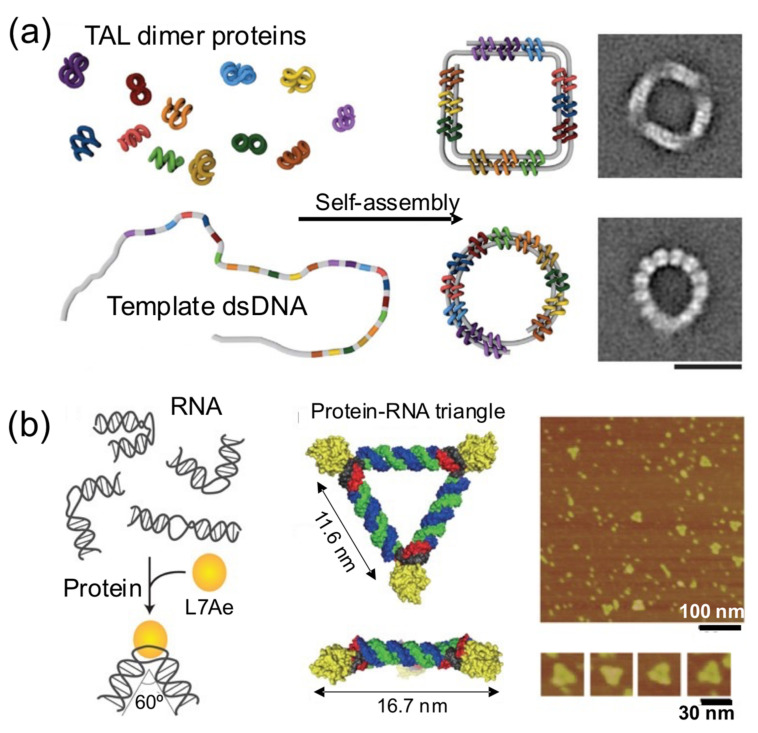
DNA is shaped by proteins into nanostructures. (**a**) Dimeric TAL effectors were programmed to attach to a dsDNA molecule in specific positions and fold it into predesigned hybrid nanostructures (left). TEM image (right). Scale bar: 20 nm. Reproduced from [17], with permission from AAAS, 2017. (**b**) An equilateral triangle is formed by bending RNA by RNA-binding protein L7Ae (left). AFM image (right). Adapted from [32], with permission from Springer Nature, 2011.

**Figure 7 nanomaterials-11-01332-f007:**
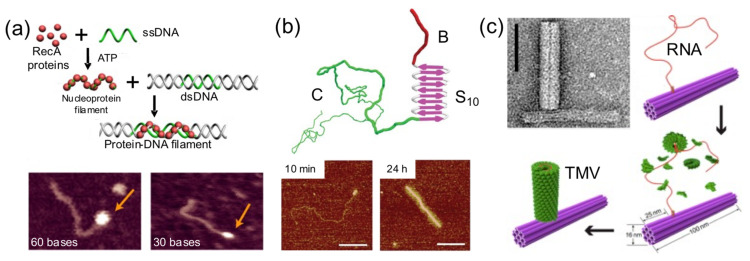
Protein self-assembly on DNA templates. (**a**) RecA protein self-assembles on a ssDNA oligonucleotide before forming complexes with dsDNA (top). AFM images of the hybrid nanostructures (bottom). Adapted from [44], with permission from American Chemical Society, 2014. (**b**) A tri-block C_4_-S_10_-B^K12^ protein polymer self-assembles onto dsDNA to form hybrid nanorods (top). AFM images (bottom). Scale bars: 100 nm. Adapted from [30], with permission from Springer Nature, 2014. (**c**) TMV capsid protein self-assembles on a ssRNA carrying the TMV origin of assembly attached to a DNA origami forming L-shaped hybrid nanostructures. TEM image (top left). Scale bar: 50 nm. Adapted from [34], with permission from American Chemical Society, 2018.

**Figure 8 nanomaterials-11-01332-f008:**
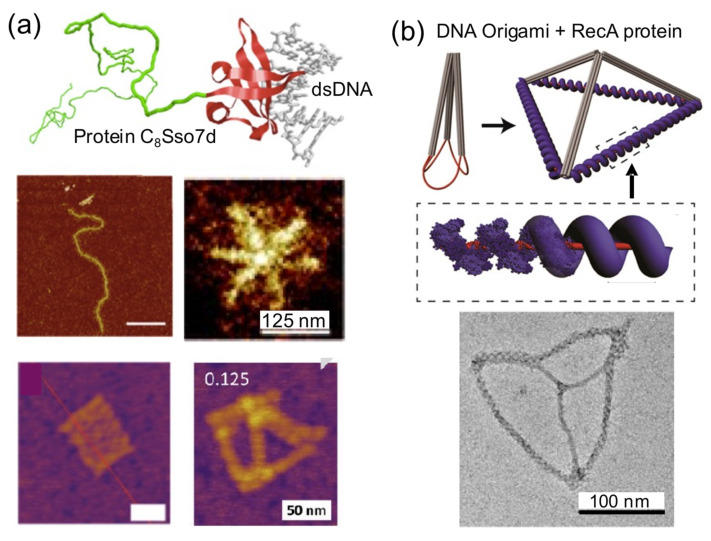
Coating and rigidizing DNA with proteins. (**a**) Diblock protein C_8_Sso7d is able to coat and rigidize dsDNA molecules into branched DNA to form hybrid nanostars and DNA origamis. AFM images (bottom). Scale bars: 500 nm (middle left), 125 nm (middle right), and 50 nm (bottom). Top, middle, and bottom left images adapted from [31], with permission from American Chemical Society, 2017. Bottom right image adapted from [45], with permission from American Chemical Society, 2017. Middle right image adapted from [18], with permission from Royal Society of Chemistry, 2019. (**b**) RecA protein self-assembles on ssDNA stretches incorporated into a DNA origami rigidizing those parts and forming a tetrahedral hybrid nanostructure. TEM image (bottom). Adapted from [50], with permission from American Chemical Society, 2017.

**Figure 9 nanomaterials-11-01332-f009:**
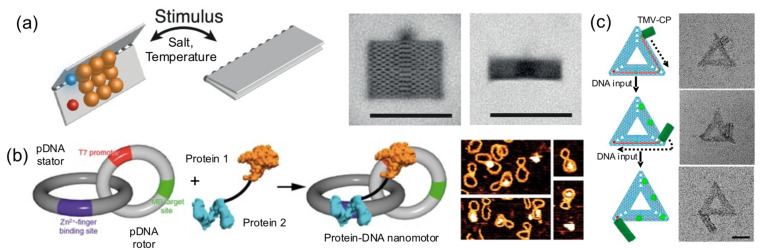
Dynamic hybrid nanostructures. (**a**) ELP opens and closes a hinge-type DNA origami upon a stimulus. TEM images (left). Scale bars: 100 nm. Adapted from [24]. (**b**) Two intertwined dsDNA rings harbored a dimerize Zinc finger and an RNA polymerase, causing the latter to rotate the ring with a T7 promotor. AFM images (right). Reprinted from [37], with permission from Springer Nature, 2018. (**c**) Dynamic assembly of Tobacco Mosaic Virus coat proteins (TMV-CP) on a DNA Origami controlled by DNA oligonucleotide toehold-mediated strand exchange. Scale bar: 50 nm. Adapted from [35], with permission from American Chemical Society, 2020.

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
