# Peer review of "Strategies to Build Hybrid Protein–DNA Nanostructures"

_nanomaterials, 2021, doi:10.3390/nano11051332_

Round 1

Reviewer 1 Report

In this paper, Armando Hernadez-Garcia reviews the field of hybrid DNA-protein nanostructures, with a focus on structural protein-DNA nanotechnology, and the fabrication methodologies used. I like the paper. It is easy to follow, and well written. It is quite short, but for me that is fine for an introduction to the topic, especially as the author is careful to define the scope. I would recommend it for publication in Nanomaterials. I include below some minor comments to help the author improve the manuscript prior to publication.

Comments…

P1… In the paragraph beginning “The scope of structural DNA nanotechnology…”, it is worth defining these as ‘spherical nucleic acids’, as that is how they are most commonly referred to in the literature.

P2 L51… what about DNAzymes? and aptamers? It is not true to say that DNA lacks catalytic activity and molecular recognition. Same for L71.

P7 L226… “Directing and organizing DNA on space” doesn’t gramatically make sense… should it be “in space”?

P12 L394… In the Perpectives section, I feel the author should expand upon the potential applications of the structures discussed in the review. It is mentioned several times that such structures will have “new and useful applicatios”, but there is currently little description of what these might be.

The figures are in general suitable and informative. I would suggest the authors revisit them to make sure all text is large enough to be read in the print version, though.

Reviewer 2 Report

This article is a decent review of protein-DNA structural nanotechnology.    Much of this material has also been reviewed by Stephanopolous (Chem, 2020), so the impact of this work will be somewhat low.  However, it is easy to read and is therefore accessible for those outside the field.

  1. It is not clear what the scope of the review intends to be (an overview, or comprehensive).  I thought the review was comprehensive until I encountered section 4.6, which opens the door to other DNA-protein assemblies that are mobile that aren't reviewed here.  The author should be more clear here about how those specific examples were chosen while others were left out.  Also, this last paragraph of this section should be included in a figure.
  2. The small text in the figures needs to be improved--in most cases it is blurry.

Reviewer 3 Report

The paper by A. Hernandez-Garcia provides a comprehensive review on recent developed strategies to build hybrid protein-DNA nanostructures. It focuses on the structural role that proteins could have when they combine with different DNA building blocks.

The review is very interesting and could be of general interest. It is well written and has a clear organization of the arguments. I suggest to add an additional chapter describing the possible functional applications of the protein-DNA nanomaterials.

Round 2

Reviewer 3 Report

none